# Approximating missing epidemiological data for cervical cancer through Footprinting: A case study in India

Irene Man*, Damien Georges, Maxime Bonjour, Iacopo Baussano

Early Detection, Prevention and Infections Branch, International Agency for Research on Cancer (IARC/WHO), Lyon, France

**Abstract** Local cervical cancer epidemiological data essential to project the context-specific impact of cervical cancer preventive measures are often missing. We developed a framework, here-after named Footprinting, to approximate missing data on sexual behaviour, human papillomavirus (HPV) prevalence, or cervical cancer incidence, and applied it to an Indian case study. With our framework, we (1) identified clusters of Indian states with similar cervical cancer incidence patterns, (2) classified states without incidence data to the identified clusters based on similarity in sexual behaviour, (3) approximated missing cervical cancer incidence and HPV prevalence data based on available data within each cluster. Two main patterns of cervical cancer incidence, characterized by high and low incidence, were identified. Based on the patterns in the sexual behaviour data, all Indian states with missing data on cervical cancer incidence were classified to the low-incidence cluster. Finally, missing data on cervical cancer incidence and HPV prevalence were approximated based on the mean of the available data within each cluster. With the Footprinting framework, we approximated missing cervical cancer epidemiological data and made context-specific impact projections for cervical cancer preventive measures, to assist public health decisions on cervical cancer prevention in India and other countries.

*For correspondence:
mani@iarc.who.int

Competing interest: The authors declare that no competing interests exist.

## Editor's evaluation

This study presents a useful framework for estimating missing data in cervical cancer epidemiology. The evidence supporting the authors' claims is solid, although validation studies in other populations will strengthen the methodology. The work will be of broad interest to researchers and policymakers interested in evaluating the impact of cervical cancer prevention measures.

## Introduction

Cervical cancer is an important source of disease burden worldwide (*de Martel et al., 2017*). In 2020, the number of new cases and deaths due to cervical cancer worldwide were estimated to be 604,000 and 342,000, respectively (*Sung et al., 2021*). Vaccination against human papillomavirus (HPV), cervical cancer screening, and treatment of pre-cancer and cancer can reduce the burden of cervical cancer (*Lei et al., 2020*; *Bouvard et al., 2021*), but access to these preventive measures is still limited in many settings, especially in low- and middle-income countries (LMICs) (*Bruni et al., 2021*; *de Sanjose and Tsu, 2019*; *Bonjour et al., 2021*). To accelerate the scale-up of cervical cancer prevention worldwide, the World Health Organization developed a global strategy to eliminate cervical cancer as a public health problem (*WHO, 2021*). The strategy proposes an elimination target of 4 cases per 100,000 women-years (age-standardized) with three intervention targets: 90% of girls vaccinated against HPV by age 15; 70% of women receiving twice-lifetime screening with high-performance

testing; and 90% of women having access to cervical pre-cancer and cancer treatment, and palliative care.

For the WHO's aspirational global targets to be perceived as realistic, achievable, and equitable, they must be adapted to local context (*Tsu, 2020*). The local need for and impact of cervical cancer preventive measures depend on the burden in a given population, which is determined by context-specific sexual behaviour, and HPV prevalence (*Guan et al., 2012*). Local data on these aspects are therefore crucial to derive projections of the health and economic impact of possible interventions. When based on adequate data, impact projections of cervical cancer preventive measures can help local health authorities set adequate public health targets and allocate resources accordingly (*Goldie et al., 2006*).

However, local epidemiological data for cervical cancer needed to derive impact projections are sometimes missing. High-quality type- and age-specific data on HPV prevalence and cervical cancer incidence from local populations are often unavailable. The same holds for adequate data on sexual behaviour, for example, data on sex outside marriage, which are also prone to bias, for example, social desirability and recall bias (*Morris et al., 2014*; *Kelly et al., 2013*). When essential epidemiological data for projections are missing, there are two main possible solutions: collection and approximation. Collection of new data in a local context would be the preferred option. However, this could be time- and resource-demanding and therefore not always feasible. Alternatively, missing data on a given population can be approximated using available data from populations sharing similar characteristics.

In this paper, we propose a framework, hereafter named Footprinting, to approximate missing cervical cancer epidemiological data for a selected number of geographical units within a larger geographical target area, to derive impact projections of cervical cancer prevention. The framework is presented using a case study in India, the country with the world's highest expected burden of cervical cancer (*Bonjour et al., 2021*) and very limited access to cervical cancer preventive measures (*Sankaranarayanan et al., 2019*). To assist local public health decision-making in India, we applied Footprinting to approximate missing Indian state-specific cervical cancer incidence and HPV prevalence data and so to enable impact projections of cervical cancer preventive measures with state-specific granularity.

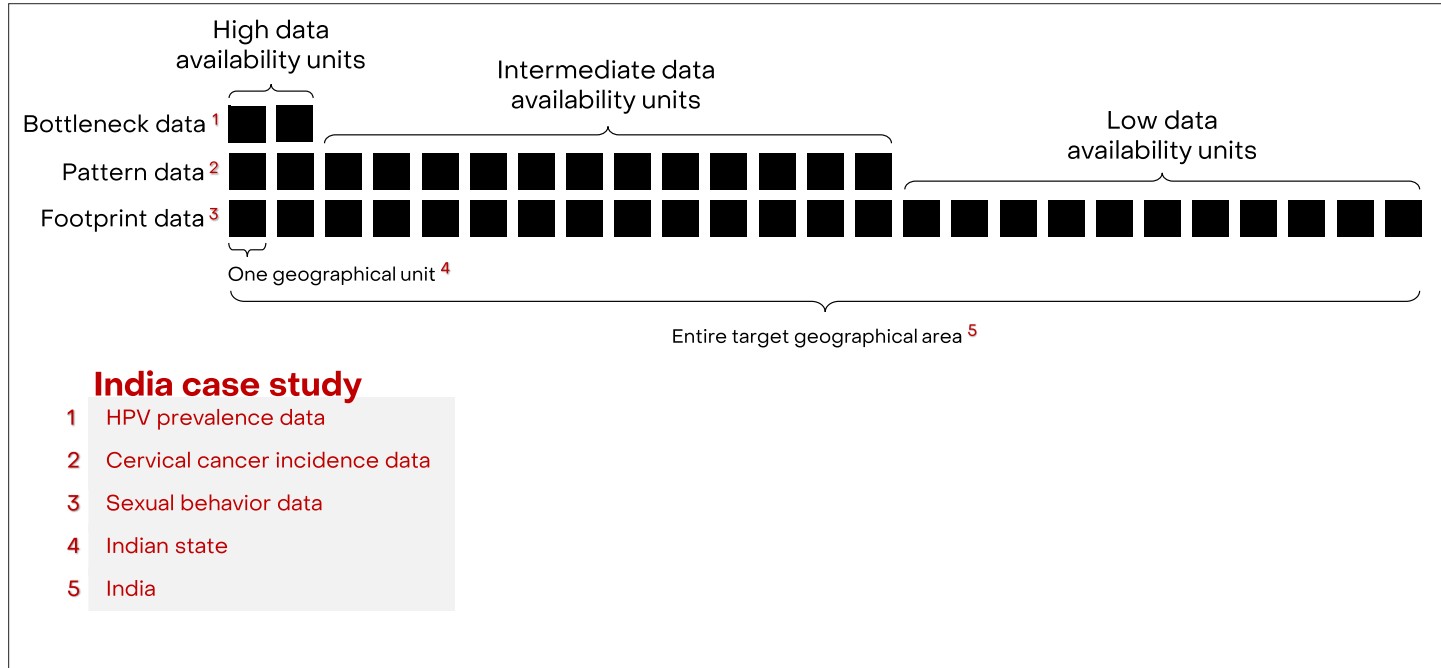

**Figure 1.** Hierarchical structure of availability of cervical cancer epidemiological data.

The online version of this article includes the following source data for figure 1:

**Source data 1.** Overview of availability of cervical cancer epidemiological data by state.

## Results

### Footprinting framework

We developed a framework, Footprinting, to approximate missing data on the three key aspects of cervical cancer epidemiology: sexual behaviour, HPV prevalence, and cervical cancer incidence. For convenience of explaining the framework, missing data across geographical units are for the moment assumed to occur in a hierarchical manner, that is, geographical units are ordered according to their levels of data availability (*Figure 1*). At the highest level, there are a small number of geographic units for which data on all three key aspects are available. For the Indian case study, there were two such states (out of 25 states or groups of states) with the high-quality type- and age-specific HPV prevalence data needed for impact projections (*Franceschi et al., 2005*; *Dutta et al., 2012*; *Kataria et al., 2022*). The remaining geographic units are further divided into levels of intermediate and low data availability. In the Indian case study, there were 12 states with cervical cancer incidence (*Bray et al., 2017*; *Report of National Cancer Registry Programme, 2020*) and sexual behaviour data, which were assigned to the intermediate level of data availability. The remaining 11 states only had data on sexual behaviour (*National Behavioural Surveillance Survey: General Population, 2006*) and were assigned to the low level of data availability. The three data sources with increasing data availability are labelled as 'Bottleneck', 'Pattern', and 'Footprint' data (*Figure 1*). See *Figure 1—source data 1* for the definition of the 25 states and a detailed overview of data availability by state.

To address the hierarchical form of missing data, we propose a three-step approach labelled as 'Clustering', 'Classification', and 'Projection'. In brief, the approach first identifies clusters of geographical units sharing similar patterns of cervical cancer epidemiology and then uses the available data within each cluster to approximate data and extrapolate impact projections to geographical units with lower data availability. The details of the respective steps are as follows:

1.  Clustering step

    In the Clustering step, clusters of geographical units sharing similar patterns of cancer epidemiology are identified. This step corresponds to unsupervised learning in machine learning terminology (*James et al., 2013*). Clustering should be done based on a source of Pattern data that has large enough coverage over the all geographical units. In the Indian case study, cervical cancer incidence data were available in 14 out of 25 Indian states and was therefore suitable. As a constraint, each of the resulting clusters must contain at least one state with high data availability. This ensures that each Indian state with an intermediate level of data availability is matched with a state with high level of data availability, which is needed to approximate missing data.

2.  Classification step

    In the second step, geographical units with the lowest level of data availability, which have not yet been clustered, are then classified into the identified clusters. This step corresponds to supervised learning in machine learning terminology (*James et al., 2013*). Classification is based on the similarity between geographical units according to the Footprint data, which should be available for the remaining unclustered geographical units. In the Indian case study, sexual behaviour data would be suitable. As in the Clustering step, the classification step matches each Indian state with the lowest level of data availability to states with higher levels of data availability within the same cluster, in order to approximate the missing cervical cancer incidence and HPV prevalence data.

3.  Projection step

    In the last step, missing data are approximated based on available data from other geographical units within the same cluster, for example, based on the mean or median of the available data. If the Classification step also provided the probability of belonging to each cluster, approximation could even be based on weighted averages of different clusters. With the approximated data, it is then possible to calibrate projection models, that is, HPV transmission and cervical cancer progression models, and derive context-specific impact projections for each geographical unit. Alternatively, as a less computationally demanding approach, it is also possible to calibrate

projection models for the geographical units of the highest level of data availability only, and then scale the projections to the other geographical units within each cluster.

As previously mentioned, we assumed that data availability occurs hierarchically. However, the framework can also be applied with less stringent data requirements. Firstly, the source of Footprint data does not necessarily need to cover all geographical units. It is possible to train a classifier in the classification step with Footprint data available for only a part of clustered geographical units. Secondly, if none of the key cervical cancer epidemiological data (sexual behaviour, HPV prevalence, and cervical cancer incidence data) have large enough coverage to serve as Footprint data, alternative indicators of similarity, such as human development index and geographical distance, could also be used as substitutes. However, this might result in suboptimal classification, as we expect these indicators to correlate less well with cervical cancer risk. Finally, for the projection step, data on cervical cancer incidence, sexual behaviour, and HPV prevalence needed to calibrate projection models do not necessarily need to belong to the same geographical unit. Calibration can be performed as long as the three types of data are available within each cluster.

With these less stringent data requirements, the proposed framework should be sufficiently flexible to be applied to many situations. However, one should still be cautious in applying the framework when little data are available. If the data are not sufficiently granular, one might need to exclude geographical units with insufficient data or redefine bigger geographical units. Furthermore, one should assess the goodness-of-fit of the obtained clustering, performance of classification, correlation of data within different clusters, and calibration fits to ensure the validity of the final impact projections.

## Clustering of cervical cancer incidence patterns in the Indian case study

As some Indian states have multiple cancer registries, we first obtained clusters of registry-specific cervical cancer incidence. See Materials and methods for the description of the source of cervical cancer incidence data and the statistical method used for clustering. Registry-specific cervical cancer incidence were obtained for up to four prefixed clusters (*Figure 2*). Model fit improved substantially when increasing the number of prefixed clusters from two to three, with the Bayesian information criterion (BIC) reducing from 6933 to 5700 (*Table 1*). Further increases in the number of prefixed clusters to four only led to marginal improvement in model fit, with a small reduction in BIC from 5700 to 5532, and a poorly defined cluster of only one registry. With the number of prefixed clusters set at five, the clustering method no longer converged. We concluded that two or three clusters were fitting to describe the patterns of cervical cancer incidence in the available data.

The cervical cancer incidence clusters differed in terms of magnitude of incidence and location of maximum incidence (*Figure 2*). When allowing two clusters, cluster 1 had a low maximum incidence of 47 cases per 100,000 women-years at age group 60–64 years, compared to cluster 2 with its higher maximum incidence of 91 cases per 100,000 women-years at the earlier age group of 55–59 years (*Figure 2*, *Table 1*). When allowing three clusters, we observed an additional cluster characterized by intermediate maximum incidence of 64 cases per 100,000 women-years at age group 60–64 years (*Figure 2*, *Table 1*). This third cluster mainly consisted of registries that had previously been assigned to the low-incidence cluster, that is, cluster 1 of the 2-clustering, while having a relatively high incidence (*Figure 2*, *Table 2*). See *Figure 2—source data 2* for additional details of the obtained clusters.

The registry clusters were then used to derive clusters of Indian states based on the majority rule (*Table 2*). When using the 2-clustering of registries, none of the states were exclusively attributed to cluster 2, hence 2-clustering could not be used for the classification step. When using 3-clustering, again none of the states were exclusively attributed to cluster 2, however, 8 and 4 states were assigned to clusters 1 and 3, that is, the clusters with low and intermediate incidence, respectively. Hence, we combined clusters 2 and 3, that is, the clusters with high and intermediate incidence as the new 'high-incidence' cluster, while keeping cluster 1 as the 'low-incidence' cluster. We note that, with these newly defined clusters, each cluster still contained at least one state with the highest level of data availability, which was necessary for the projection step. However, with the new definition of clusters, we could no longer distinguish patterns of early or late peak of incidence.

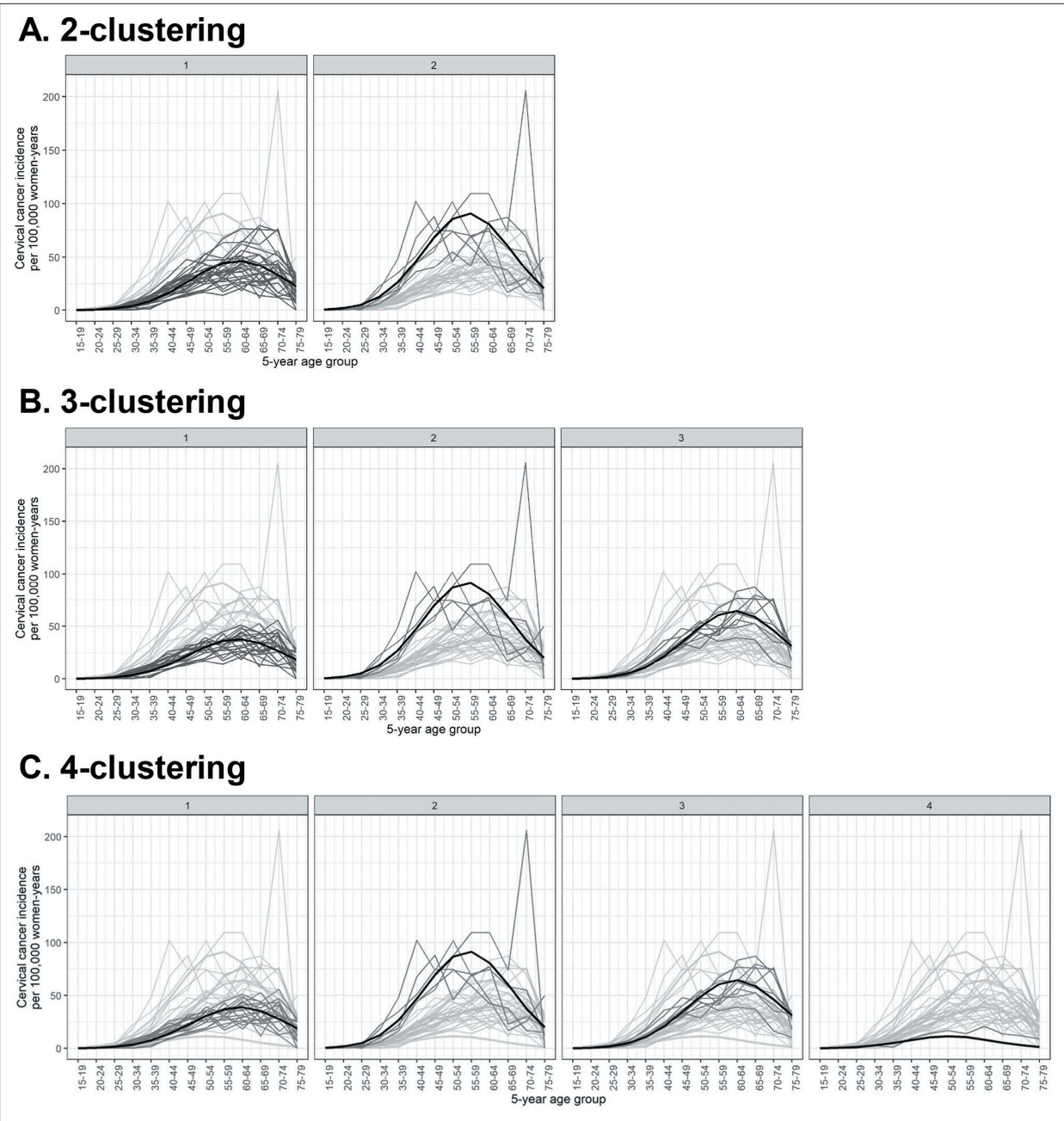

**Figure 2.** Identified clusters of registry-specific cervical cancer incidence. Clusterings under (**A**) 2, (**B**) 3, and (**C**) 4 prefixed clusters. Each panel within a row corresponds to a cluster within a *k*-clustering, with the cluster label given on top of the panel. The cervical cancer incidence data were extracted from volume XI of Cancer Incidence in Five Continents (CI5) (***Bray et al., 2017***) and the 2012–2016 report by the Indian National Centre for Disease Informatics and Research (NCDIR) (***Report of National Cancer Registry Programme, 2020***). Black: cluster mean of cervical cancer incidence; dark grey: registry incidence assigned to the cluster; light grey: registry incidence assigned to other clusters.

The online version of this article includes the following source data and figure supplement(s) for figure 2:

**Source data 1.** Registry-specific cervical cancer incidence data from Cancer Incidence in Five Continents (CI5) and National Centre for Disease Informatics and Research (NCDIR).

**Source data 2.** Estimated model parameters under Poisson regression models.

**Figure supplement 1.** Registry-specific cervical cancer incidence data from Cancer Incidence in Five Continents (CI5) and National Centre for Disease Informatics and Research (NCDIR).

**Figure supplement 2.** Mean age-specific cervical cancer incidence by cluster.

**Table 1.** Estimated parameters of clusters of cervical cancer incidence patterns.

| Number of prefixed clusters | BIC* | Cluster label $i$ | Number (%) of registries in cluster | Maximum incidence† | Maximum incidence pattern | Maximum incidence age group‡ | Maximum incidence age group pattern |
|---|---|---|---|---|---|---|---|
| | | 1 | 27 (82%) | 47 cases | Low | 60–64 years | Late |
| 2 | 6933 | 2 | 6 (18%) | 91 cases | High | 55–59 years | Early |
| | | 1 | 19 (58%) | 38 cases | Low | 60–64 years | Late |
| | | 2 | 5 (15%) | 92 cases | High | 55–59 years | Early |
| 3 | 5700 | 3 | 9 (27%) | 64 cases | Intermediate | 60–64 years | Late |
| | | 1 | 18 (55%) | 39 cases | Low | 60–64 years | Late |
| | | 2 | 5 (15%) | 92 cases | High | 55–59 years | Early |
| | | 3 | 9 (27%) | 64 cases | Intermediate | 60–64 years | Late |
| 4 | 5532 | 4 | 1 (3%) | 20 cases | Very low | 60–64 years | Early |

*Bayesian information criterion for evaluating the goodness-of-fit of obtained clustering.

†Maximum incidence given in cases per 100,000 women-years.

‡Five-year age group in which the maximum incidence is located.

## Classification of cervical cancer incidence patterns based on sexual behaviour data in the Indian case study

A random forest (RF) classifier was constructed using sexual behaviour data corresponding to the states with identified clusters. See Materials and methods for the details of the source of sexual behaviour data, the variables included, and the statistical method used for classification. The variables with the first, second, and third highest predictive values for cluster of cervical cancer incidence patterns were *'proportion of urban male respondents reporting sex with non-regular partners in the last 12 months'*, *'median age of first sex in rural males'*, and *'median age of first sex in urban females'*, respectively (*Figure 3—source data 2*). In particular, there was a good distinction between the high- and low-incidence clusters in terms of *'proportion of urban male respondents reporting sex with non-regular partners in the last 12 months'*, with high proportions associated with the high-incidence cluster (*Figure 3*). High values of *'median age of first sex in females'* and low values of *'median age of first age in males'* were also associated for the high-incidence cluster, although the distinction was less clear.

The estimated out-of-bag error of the constructed classifier was 29%. When applying the constructed classifier to the Indian states with identified clusters, only Karnataka and other North Eastern states (2 of 14 states; 14%) were wrongly classified to the low-incidence cluster (*Table 3*). Visualization shows that the sexual behaviour data for these two states resemble the other states belonging to the low-incidence cluster, despite being clustered into the high-incidence cluster (*Figure 3*, *Figure 3—figure supplement 1*).

Subsequently, the classifier was applied to classify the remaining states without cervical cancer incidence data and thus with unknown cluster. All 11 remaining Indian states received a higher probability of belonging to the low-incidence cluster (*Table 3*). Indeed, *Figure 3* shows that the sexual behaviour data of the states with unknown cluster (indicated in grey) were generally closer to the sexual behaviour data of the states of the low-incidence cluster (indicated in red) than those of the high-incidence cluster (indicated in blue). Hence, we identified in total 19 and 6 states for the low- and high-incidence clusters, respectively.

Finally, missing cervical cancer incidence data and HPV prevalence were approximated based on the mean within each cluster (*Figure 2—figure supplement 2*). Approximation of HPV prevalence was based on the only one prevalence survey we could identify per cluster (*Franceschi et al., 2005*; *Dutta et al., 2012*). We verified that the HPV prevalence reported by the survey corresponding to the high-incidence cluster was higher than the prevalence reported by the one corresponding to the low-incidence cluster: HPV prevalence of 16.9% vs 9.8% (in women in the age range 20–60 years). This 1.7-fold difference in HPV prevalence was in the same order of magnitude as the 1.9-fold difference we

**Table 2.** Clustering of cervical cancer incidence of Indian states based on clustering of registries.

| State/group of states* | 2-Clustering | | 3-Clustering | | | 4-Clustering | | | |
|---|---|---|---|---|---|---|---|---|---|
| | 1 (low, late) | 2 (high, early) | 1 (low, late) | 2 (high, early) | 3 (interm., late) | 1 (low, late) | 2 (high, early) | 3 (interm., late) | 4 (very low, early) |
| Andhra Pradesh | ● | | ● | | | ● | | | |
| Assam | ●●● | | ●●● | | | ●● | | | ● |
| Delhi | ● | | | | ● | | | ● | |
| Gujarat+Dadra and Nagar Haveli | ● | | ● | | | ● | | | |
| Karnataka | ● | | | | ● | | | ● | |
| Kerala +Lakshadweep | ●● | | ●● | | | ●● | | | |
| Madhya Pradesh | ● | | | | ● | | | ● | |
| Maharashtra | ●●●●●● | ● | ●●●● | | ●●● | ●●●● | | ●●● | |
| Manipur | ●● | | ●● | | | ●● | | | |
| Other North Eastern states† | ●●●●● | ●●●● | ●●●● | ●●●● | ● | ●●●● | ●●●● | ● | |
| Punjab +Chandigarh | ● | | | | ● | | | ● | |
| Sikkim | ● | | ● | | | ● | | | |
| Tamil Nadu +Puducherry | | ● | | ● | | | ● | | |
| West Bengal +Andaman and Nicobar Islands | ● | | ● | | | ● | | | |

Each circle represents the count of one registry being assigned to the corresponding cluster. Grey shading represents the cluster including the highest number of registries, either exclusively or in a draw with another cluster.

Cluster labels and the corresponding patterns of maximum incidence and maximum incidence age group given in the second row were defined in the third, sixth, and eighth columns of **Table 1**, respectively.

*States/or groups of states were defined as reported in the 2006 National Behaviour Surveillance Survey of the National AIDS Control Organization of India (**National Behavioural Surveillance Survey: General Population, 2006**).

†Other North Eastern states included Arunachal Pradesh, Nagaland, Meghalaya, Mizoram, and Tripura.

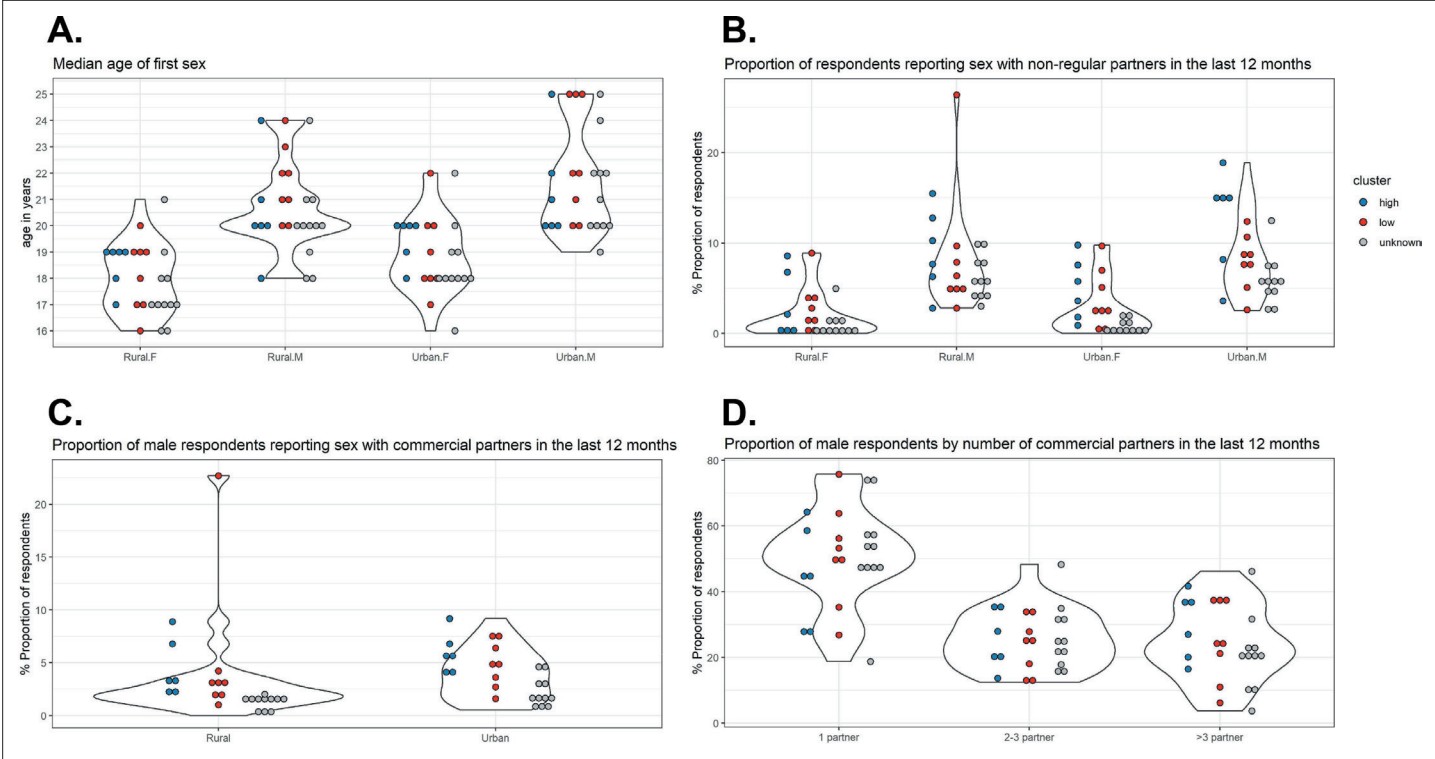

**Figure 3.** Sexual behaviour data from National AIDS Control Organization (NACO) by Indian state. Indian state-specific data on (**A**) median age of first sex, (**B**) proportion of respondents reporting sex with non-regular partners in the last 12 months, (**C**) proportion of male respondents reporting sex with commercial partners in the last 12 months, and (**D**) proportion of male respondents by number of commercial partners in the last 12 months. Each violin plot and the associated cloud of circles correspond to a sexual behaviour variable. Each circle corresponds to the data of a state (or group of states). The data were extracted from the 2006 National Behaviour Surveillance Survey of the National AIDS Control Organization of India (*National Behavioural Surveillance Survey: General Population, 2006*). Blue and red: Indian states identified in the high and low cervical cancer incidence clusters. Grey: states without cervical cancer incidence data and therefore unknown cluster.

The online version of this article includes the following source data and figure supplement(s) for figure 3:

**Source data 1.** Indian state-specific sexual behaviour data from National AIDS Control Organization (NACO).

**Source data 2.** Predictive values of the sexual behavior variables for cervical cancer incidence cluster.

**Figure supplement 1.** Indian state-specific sexual behaviour data from National AIDS Control Organization (NACO).

found for the age-standardized cervical cancer incidence between the two clusters (17.9 vs 9.01 cases per 100,000 women-years). The final step of deriving impact projections for cervical cancer preventive measures for the whole of India with state-specific granularity was reported elsewhere (*de Carvalho et al., 2023*; *Man et al., 2022*).

## Discussion

In this paper, we developed the Footprinting framework to approximate missing cervical cancer epidemiological data in some geographical units when deriving impact projections of cervical cancer preventive measures for a larger geographical area. In brief, the framework identified clusters of geographical units sharing similar patterns of cervical cancer epidemiology and uses the available data within each cluster to approximate data and extrapolate impact projections to geographical units with lower data availability. The framework was demonstrated using a case study approximating missing cervical cancer incidence and HPV prevalence data for a selection of Indian states. With the framework, we have derived, for the first time, impact projections of cervical cancer preventive measures for the whole of India with state-specific granularity (*de Carvalho et al., 2023*; *Man et al., 2022*).

This work has also generated a better understanding of cervical cancer epidemiology across India. We found that India can be divided into two main groups of 19 and 6 Indian states or groups of states

**Table 3.** Identified and classified cluster of cervical cancer incidence pattern by Indian state.

| Cervical cancer incidence data* | State/group of states† | Identified cluster‡ | Classified cluster§ | Probability of belonging to the low-incidence cluster |
|---|---|---|---|---|
| | Andhra Pradesh | Low | Low | 0.60 |
| | Assam | Low | Low | 0.69 |
| | Delhi | High | High | 0.42 |
| | Gujarat+Dadra and Nagar Haveli | Low | Low | 0.69 |
| | Karnataka | High | Low | 0.63 |
| | Kerala+Lakshadweep | Low | Low | 0.60 |
| | Madhya Pradesh | High | High | 0.44 |
| | Maharashtra | Low | Low | 0.57 |
| | Manipur | Low | Low | 0.65 |
| | Other North Eastern states¶ | High | Low | 0.53 |
| | Punjab+Chandigarh | High | High | 0.41 |
| | Sikkim | Low | Low | 0.63 |
| | Tamil Nadu+Puducherry | High | High | 0.38 |
| Available | West Bengal+Andaman and Nicobar Islands | Low | Low | 0.71 |
| | Bihar | - | Low | 0.67 |
| | Chhattisgarh | - | Low | 0.66 |
| | Goa+Daman and Diu | - | Low | 0.54 |
| | Haryana | - | Low | 0.66 |
| | Himachal Pradesh | - | Low | 0.58 |
| | Jammu and Kashmir | - | Low | 0.63 |
| | Jharkhand | - | Low | 0.71 |
| | Orissa | - | Low | 0.68 |
| | Rajasthan | - | Low | 0.66 |
| | Uttar Pradesh | - | Low | 0.64 |
| Unavailable | Uttarakhand | - | Low | 0.69 |

*Availability of cervical cancer incidence data was based on the incidence data from volume XI of Cancer Incidence in Five Continents (CI5) (**Bray et al., 2017**) and the 2012–2016 report of the National Centre for Disease Informatics and Research (NCDIR) (**Report of National Cancer Registry Programme, 2020**).

†States/groups of states were defined as reported in the 2006 National Behaviour Surveillance Survey of the National AIDS Control Organization of India (**National Behavioural Surveillance Survey: General Population, 2006**).

‡Identified clusters derived in the Clustering step.

§Classified clusters derived in the Classification step. A given state was classified to the low-incidence cluster if the probability of belonging to the low-incidence cluster (given in the next column) was above 0.50. For the Indian states with available cervical cancer incidence data and hence already in an identified cluster, classification was done for the purpose of validation.

¶Other North Eastern states included Arunachal Pradesh, Nagaland, Meghalaya, Mizoram, and Tripura.

that are characterized by low or high cervical cancer incidence, respectively. As expected, and in line with previous studies, individuals, in particular men, in high-incidence states had more sexual activity with non-regular partners, including commercial partners, than in low-incidence states (**Vaccarella et al., 2006**; **Schulte-Frohlinde et al., 2022**). While early sexual debut in women has also previously

been suggested to be associated with high cervical cancer incidence and HPV positivity (*Vaccarella et al., 2006*; *Schulte-Frohlinde et al., 2022*), it was associated with lower cervical cancer incidence in the dataset we considered. We hypothesize that, for the Indian context, early sexual debut is common in states with a larger rural population, among whom less sexual activity occurs with non-regular partners, which is the main determining factor for a lower risk of cervical cancer. With the urbanization of rural areas, which often entails evolving socio-cultural norms, it is possible that more Indian states may shift to a high cancer incidence pattern, with an accompanying early peak in incidence (*Baussano et al., 2016*).

In our analysis of classifying clusters of cervical cancer incidence, we only included some of the sexual behaviour variables available in the NACO report (*National Behavioural Surveillance Survey: General Population, 2006*). We selected variables that were previously shown to be risk factors of HPV infection and cervical cancer risk and that are commonly available (e.g. age of sexual debut and number of sexual partners) so that the analyses can be easily applied to other settings (*Vaccarella et al., 2006*; *Schulte-Frohlinde et al., 2022*). In the Indian case study, the good classification performance shows that using the selected set was sufficient. As sexual behaviour variables are highly correlated, adding more variables might cause overfitting.

It should be noted that our Footprinting framework is similar to other extrapolation approaches previously used in model-based projection studies targeting large geographical areas with missing data, for example, a collection of LMICs or European countries (*Brisson et al., 2020*; *Qendri et al., 2020*). While the rationale behind different extrapolation approaches is similar, which is to approximate missing data from other geographical units with similar epidemiological indicators, there are also differences. A strength of our framework is that it relies on the observed patterns of epidemiology in the data to select key geographical units from which impact projections are extrapolated to other units instead of working with a predefined selection of key units. This allows the selection of key units that maximizes the representation of different epidemiological patterns in the data. Moreover, it also helps to pinpoint geographical units that could be interesting for future data collection efforts. Secondly, we used a newly developed clustering method (*Subtil et al., 2017*; *Klich et al., 2021*) that is able to assess the similarities between cervical cancer incidence of different geographical units based on the entire age-specific pattern, instead of clustering by age-standardized cervical cancer incidence or cervical cancer incidence in a certain age group only. Finally, we provided a detailed description of the clustering/mapping steps and intermediate results, which make them more reproducible and falsifiable.

Our application of Footprinting on the Indian case study also bears some resemblance with the extrapolation of cervical cancer incidence by GLOBOCAN's nationwide estimates of cervical cancer incidence in India (*Bray et al., 2017*). Essentially, in GLOBOCAN, missing incidence was extrapolated based on urban or rural residency as a footprint, while we used sexual behaviour for this purpose and considered each state separately, which is necessary for state-specific impact projections. As a result, we neglected the variation between rural and urban areas within Indian states, which is a limitation of our analysis. We expect that Footprinting with further stratification of states by rural/urban residency could improve the approximation. Furthermore, our nationwide estimate of cervical cancer incidence derived from aggregating the state-specific estimates (reported in a separate manuscript; *Man et al., 2022*) was lower than the estimate reported by GLOBOCAN. This could be explained by the use of different methods of extrapolation and the fact that we included data from 17 additional cancer registries with relatively low incidence not included in GLOBOCAN estimates. Various possible adaptations of the proposed Footprinting framework are worth mentioning. Firstly, in the suboptimal situation where none of the relevant cervical cancer epidemiology data are available in some geographical units, data on indicators of human development and geographical location could be used as Footprint data. Secondly, while we focused on epidemiological data for cervical cancer, Footprinting could be used to approximate missing economic data (e.g. treatment or vaccine delivery costs) that are needed to assess the health economic impact of cervical cancer preventive measures, given that relevant Footprint data can be defined and collected. It is important to note that, in general, the applicability the proposed framework depends on the amount of data available. However, in our opinion, lack of data is a general challenge for approximating missing data, rather than a weakness particular to our methodology. By allowing possible adaptations, we believe that our framework is sufficiently flexible to effectively address missing data in many situations.

This work has provided a comprehensive framework to dealing with the important and ubiquitous challenge of missing data on cervical cancer epidemiology. By using the proposed framework, it is possible to derive robust and context-specific impact projections for cervical cancer preventive measures for a wide range of geographical settings. Such projections can assist local health authorities to plan and implement cervical cancer preventive strategies that are adapted to local needs and resources, intensifying efforts to reduce the high burden of cervical cancer still existing in many countries in low-resource settings.

# Materials and methods

## Data sources

In this section, we describe the data sources used in the Indian case study. The primary source of cervical cancer incidence data, which was used as Pattern data in the Clustering step, was cancer registry data from volume XI of Cancer Incidence in Five Continents (CI5) (*Bray et al., 2017*). It comprised incidence data from 16 cancer registries in 10 of the 25 Indian states (some states had more than one registry). In addition, cervical cancer incidence data were extracted from the 2012–2016 report by the Indian National Centre for Disease Informatics and Research (NCDIR) to provide data from 17 additional cancer registries not included in CI5 (*Report of National Cancer Registry Programme, 2020*). When data of a registry is both were reported both in CI5 and NCDIR, we only used the data from CI5. Combining the two sources provided incidence data for 33 registries in 14 Indian states. Cervical cancer incidence was reported by number of cases per 100,000 women-years, stratified by 5-year age groups from 15 to 79 years. See *Figure 2—figure supplement 1* and *Figure 2—source data 1* for the extracted incidence data by state.

Sexual behaviour data, which were used as Footprint data in the Classification step, were from the report of the National Behaviour Surveillance Survey by the National AIDS Control Organization (NACO) of India in 2006, which was the most recent edition at the moment of writing (*National Behavioural Surveillance Survey: General Population, 2006*). Data for all 25 Indian states were available in the survey. Sexual behaviour data by Indian state in the form of aggregate statistics of survey respondents were available for the following 4 groups of 12 variables:

- *Median age of first sex* – stratified by residence (*urban/rural*) and sex (*male/female*), resulting in four variables.
- *Proportion of respondents reporting sex with non-regular partners in the last 12* months – stratified by residence (*urban/rural*) and sex (*male/female*), resulting in four variables.
- *Proportion of male respondents reporting sex with commercial partners in the last 12* months – stratified by residence (*urban/rural*), resulting in two variables.
- *Proportion of male respondents by number of commercial partners in the last 12* months – restricted to respondents with at least one commercial partner and divided into three categories (*1/2–3/>3*). As the three proportions always sum up to one and are therefore correlated, we omitted one category, resulting in two variables.

See *Figure 3—figure supplement 1* and *Figure 3—source data 1* for the extracted sexual behaviour data by state.

## Method to cluster cervical cancer incidence patterns

The statistical method employed in the Clustering step to cluster registry-specific cervical cancer incidence data was a Poisson-regression-based CEM clustering algorithm (*Subtil et al., 2017*; *Klich et al., 2021*), described in detail in Appendix 1. Briefly, clusters of age-specific cervical cancer incidence were obtained by likelihood-based optimization under Poisson regression model. The Poisson regression model for each cluster was characterized by three parameters: an intercept, one parameter for age, and one for the square of age. This parametric form was chosen to match the general pattern of incidence by age, namely, increasing from zero incidence from the youngest age group, then decreasing after reaching the maximum incidence (*Figure 2—figure supplement 1*). Application of the clustering method required prefixing the number of clusters $k$. The goodness-of-fit of each $k$-clustering was evaluated based on the BIC. To transform the obtained clustering of registry-specific data to clustering of Indian states for states with multiple registries, we assigned each state to the cluster that included the highest number of registries, that is, according to a majority rule.

## Method to classify cervical cancer incidence patterns based on sexual behaviour data

In the Classification step, we assigned the remaining states without cervical incidence data to the identified clusters based on RF using the sexual behaviour data as Footprint data (*Breiman, 2001*). The RF classifier was constructed using sexual behaviour data from states with identified clusters. The predictive value of each variable was evaluated with the mean decrease in accuracy, which expressed how much the accuracy of the model decreased if the variable was excluded. The performance of the classification step was validated by both out-of-bag error estimate and by applying the constructed classifier to the sexual behaviour data from states with identified clusters. Subsequently, the constructed classifier was applied to the sexual behaviour data from states without identified clusters, providing the probability to belong to each cluster. Each state was classified to the cluster receiving the highest probability. Classification was performed using the R package *party* version 1.3–7 with the following setting: *cforest_control(teststat = 'quad', testtype = 'Univariate', mincriterion = 0.9, ntree = 50 000,, mtry = 3, maxdepth = 2, minsplit = 0, minbucket = 0)*.

Finally, in the projection step, missing data on cervical cancer incidence and HPV prevalence were approximated based on the mean within each cluster. Results were validated based on the ratios of HPV prevalence and cervical cancer incidence across clusters. Derivation of impact projections was reported elsewhere (*de Carvalho et al., 2023*; *Man et al., 2022*).

## Acknowledgements

This study was funded by the Bill and Melinda Gates Foundation (grant numbers: OPP48979; INV-039876). The funder had no role in study design, data collection and analysis, decision to publish, or preparation of the manuscript. For the authors identified as personnel of the International Agency for Research on Cancer or World Health Organization, the authors alone are responsible for the views expressed in this article and they do not necessarily represent the decisions, policy, or views of the International Agency for Research on Cancer or World Health Organization. The designations used and the presentation of the material in this article do not imply the expression of any opinion whatsoever on the part of WHO and the IARC about the legal status of any country, territory, city, or area, or of its authorities, or concerning the delimitation of its frontiers or boundaries.

## Additional information

### Funding

| Funder | Grant reference number | Author |
|---|---|---|
| Bill and Melinda Gates Foundation | OPP48979 | Iacopo Baussano |
| Bill and Melinda Gates Foundation | INV-039876 | Iacopo Baussano |

The funders had no role in study design, data collection and interpretation, or the decision to submit the work for publication.

### Author contributions

Irene Man, Conceptualization, Data curation, Formal analysis, Investigation, Visualization, Methodology, Writing - original draft, Writing – review and editing; Damien Georges, Conceptualization, Data curation, Formal analysis, Investigation, Visualization, Methodology, Writing – review and editing; Maxime Bonjour, Software, Validation, Methodology, Writing – review and editing; Iacopo Baussano, Conceptualization, Resources, Formal analysis, Supervision, Validation, Investigation, Methodology, Writing – review and editing

### Author ORCIDs

Irene Man ⓘ http://orcid.org/0000-0003-3177-6904
Iacopo Baussano ⓘ http://orcid.org/0000-0002-7322-1862

Decision letter and Author response
Decision letter https://doi.org/10.7554/eLife.81752.sa1
Author response https://doi.org/10.7554/eLife.81752.sa2

## Additional files

### Supplementary files
• Supplementary file 1. Appendix 1 - Poisson-regression-based CEM clustering algorithm.
• MDAR checklist

### Data availability
All data used in the present study were openly available and extracted from http://ci5.iarc.fr for the cervical cancer incidence data published by the International Agency for Research on Cancer, from https://www.ncdirindia.org/All_Reports/Report_2020/resources/NCRP_2020_2012_16.pdf for the cervical cancer incidence data published by the National Centre for Disease Informatics and Research of India, and from https://www.aidsdatahub.org/sites/default/files/resource/national-bss-general-population-india-2006.pdf for the sexual behavior data published by the National AIDS Control Organisation Ministry of Health and Family Welfare Government of India. The extracted cervical cancer incidence and sexual behavior data are provided in *Figure 2—source data 1* and *Figure 3—source data 1*, respectively. The computer code regarding the Poisson-regression-based CEM clustering algorithm is available upon reasonable request to the authors. The Random forest analysis was done with the open-source R packages party available at https://cran.r-project.org/web/packages/party/index.html (*Hothorn et al., 2023*).

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
