## [Editor Report]

This study presents a useful framework for estimating missing data in cervical cancer epidemiology. The evidence supporting the authors' claims is solid, although validation studies in other populations will strengthen the methodology. The work will be of broad interest to researchers and policymakers interested in evaluating the impact of cervical cancer prevention measures.

---

## [Decision Letter]

**Decision letter after peer review:**

Thank you for submitting your article "Footprinting" missing epidemiological data for cervical cancer: a case study in India" for consideration by *eLife*. Your article has been reviewed by 2 peer reviewers, including Belinda Nicolau as the Reviewing Editor and Reviewer #1, and the evaluation has been overseen by Wendy Garrett as the Senior Editor. The following individuals involved in the review of your submission have agreed to reveal their identity: Esther Roura (Reviewer #1).

Essential revisions:

Both reviewers' comments raised the question about the utility of this framework when little data are available on HPV prevalence, sexual behaviour, and cervical cancer incidence. What are the minimum data required to use this technique? What are the combinations of data that one can utilize with this methodology? Moreover, this methodology has not been validated; validation studies using data from countries where complete datasets are available (e.g., the US) will strengthen this methodology.

The authors should consider changing the name "footprinting"; as it is not intuitive in the context of the manuscript. Also, the authors should provide a more comprehensive explanation of the advantages of this technique over previous approaches, discussing its strengths and limitations.

Similarly, a more comprehensive explanation of the use of the sexual behaviour data is necessary; how did the authors decide which statistics to include in the methodology (e.g., age of first sexual intercourse, number of sexual partners lifetime)? Is there a guide to know which variables are ideal to apply in the technique?

*Reviewer #1 (Recommendations for the authors):*

I have some general comments.

The word "footprinting" is not very intuitive in the context of the manuscript. It sounds a bit strange, especially in the title; in the manuscript it's fine.

I have doubts about the utility of this framework when little data is available on HPV prevalence, sexual behavior, and cervical cancer incidence. What are the minimum data required to use this technique? What combinations of data types do you think we can do with this methodology (e.g., we only have sexual behavior but can use it with the HDI)?

Regarding sexual behavior data, how do you decide the statistics to include in the methodology (e.g., the median age of first age, number of sexual partners lifetime)? Is there a guide to know which variables are more recommended to apply in the technique?

It will be interesting to validate this methodology in other regions with less available data to confirm its potential and utility.

I also have more specific comments or questions.

When a cancer registry is available in both CI5C and NCDIR, and the information is not the same, which one do you get?

Can you explain how you decide the initial assignments in the CEM clustering algorithm (ranges of 100 repetitions)?

Considering that none of the clusterings obtained using the CEM clustering algorithm are appropriate (2, 3, or 4), have you considered another method of clustering instead of combining clusters 2 and 3?

Can you explain more extensively the differences between the previous approaches and your approach, with the strengths and limitations, and why this technique is more advantageous than the other ones?

*Reviewer #2 (Recommendations for the authors):*

The proposed framework's strength is difficult to evaluate because the steps and justification for the model variables were not clearly presented, nor were the models validated. Since the whole framework is built on one single imputation, how do the authors account for uncertainties about the estimation? Perhaps the authors could consider validating these models by simulating models using data from countries where complete datasets are available (e.g., the US).

The manuscript would be strengthened by a more explicit stepwise delimitation of how to apply this model to data. The paper would be strengthened by including evidence for the utility of estimating HPV and cervical cancer rates based on sexual behaviours.

Lastly, it seems that the impact assessment of this work has already been published. Why was the current manuscript sent for publication after the paper on impact assessment was published?

---

## [Author Response]

Essential revisions:Both reviewers' comments raised the question about the utility of this framework when little data are available on HPV prevalence, sexual behaviour, and cervical cancer incidence. What are the minimum data required to use this technique? What are the combinations of data that one can utilize with this methodology?

We thank the editor for the comments and for summarizing the main comments of the reviewers. We have tried to incorporate them as much as possible in the revised manuscript. We believe that the current version is now more complete for readers who may be interested in adapting this technique to their needs.

As the editor and reviewers have pointed out, the applicability of the proposed methodology depends on the available data. In our opinion, it is a general challenge of approximating missing data, rather than a weakness particular to our methodology. In fact, we believe that our framework is flexible to address missing data in many situations.

To answer the editor’s first question, there are three minimum data requirements. In our opinion, these requirements are reasonable and flexible enough to be fulfilled in many situations.

1) The first requirement is a data source of cervical cancer incidence, sexual behaviour, or HPV prevalence that has large enough, but not necessarily complete, coverage over all geographical units of interest. This data source, called “Pattern data” in the manuscript, is used to identify the main patterns of cervical cancer epidemiology.

2) A source of cervical cancer incidence, sexual behaviour, or HPV prevalence, or even alternative proxy as HDI and geographical location with coverage over the remaining geographical units to classify the unclustered geographical units to the identified clusters. In the manuscript, this data source is called “Footprint data”. This data source needs to cover some geographical units with identified clusters but not necessary all. Coverage on a part of the clustered geographical units should be sufficient for training the classifier used in the classification step.

3) Finally, data of cervical cancer incidence, sexual behaviour, and HPV prevalence for one geographical unit within each cluster are needed for the calibration of the projection models. However, these data do not necessarily need to come from the same geographical unit, as the data within a cluster should be similar enough.

To answer editor’s second question, any combination of two of the three key cervical cancer epidemiological data, i.e., sexual behaviour, HPV prevalence, and cervical cancer incidence data, can serve as “Pattern” and “Footprint” data. As mentioned under data requirement (2) even proxies as HDI can be used as footprint data.

While we think our framework is flexible to be applied to many situation, we would like to stress that, as a general principle, we should not try overly approximate missing data when too little data are available to approximate from. This means that sometimes we might need to exclude from the analysis geographical units, or we might need to define bigger geographical units if the data are not granular enough. Only by doing so can we ensure the quality of the approximated data and the final impact projections.

Finally, it is worth noting that there are various widely recognized sources of databases of cervical cancer epidemiological data by country:

- cervical cancer incidence from CI5,

- HPV prevalence from ICO/IARC HPV information centre,

- sexual behaviour from the Demographic and Health Surveys (DHS) Program.

Therefore, there are likely ample data for application of the framework when considering countries as geographical units. For application within countries with states/provinces/municipalities as geographical units, data availability can differ from country to country.

To clarify these points, we have added the following paragraph to the Method (lines 144-163, pages 7-8): “For convenience of explanation, we assumed earlier that data availability occurs hierarchically. However, the framework can also be applied with less stringent data requirements. First, the source of Footprint data needs not necessarily cover all geographical units. It is still possible to train a classifier in the classification step with Footprint data available for only a part of clustered geographical units. Second, if none of the key cervical cancer epidemiological data (sexual behavior, HPV prevalence, and cervical cancer incidence data) have large enough coverage to serve as Footprint data, alternatives indicators of similarity, such as human development index and geographical distance, could also be used as substitute. However, the resulting classification performance might be suboptimal, as we expect these indicators to correlate less well with cervical cancer risk. Third, for the projection step, data of cervical cancer incidence, sexual behavior, and HPV prevalence needed for calibration of projection models need not necessarily belong to the same geographical unit. Calibration can be performed as long as the three types of data are available within each cluster.

With these less stringent data requirements, the proposed framework should sufficient flexible to be applied to many situations. However, one should still be cautious in applying the framework when there are little data. This means that, in some cases, we might need to exclude from the analysis some geographical units with too little data or redefine bigger geographical units if the data are not granular enough. Furthermore, we should assess the goodness-of-fit of the obtained clustering, performance of classification, correlation of data within different clusters, and calibration fits to ensure the validity of the final impact projections.”

Moreover, this methodology has not been validated; validation studies using data from countries where complete datasets are available (e.g., the US) will strengthen this methodology.

We agree that it would be very interesting to validate this proposed methodology in other regions. Unfortunately, it was beyond the scope of this work. Currently, we are working on a project in which we try to apply footprinting to a collection of low- and middle-income countries.

The authors should consider changing the name "footprinting"; as it is not intuitive in the context of the manuscript.

We have changed the title into ‘Approximating missing epidemiological data for cervical cancer through “footprinting”: a case study in India’ to explain the purpose of “footprinting”.

Also, the authors should provide a more comprehensive explanation of the advantages of this technique over previous approaches, discussing its strengths and limitations.

To our knowledge, in the field of HPV and cervical cancer control, there are two publications on multi-country modelling for cervical cancer prevention with similar approaches (sometimes also called mapping) as the approach we proposed [ref #29 Brisson 2020. ref #30 Qendri 2020]. In these publications, similar data (sexual behaviour, cervical cancer data, and HPV prevalence) were used for clustering of countries. In our opinion (lines 329-334, page 16), an advantage of our approach is that we base our choice of key geographical units from which impact projections are extrapolated from on the pattern discovered in the data. Other approaches work with prefixed key geographical units because projection models have been calibrated to these geographical units in previous publications. However, no formal analyses have been done to show how well these key geographical units represent the cervical cancer epidemiological patterns across the geographical area of interest.

In addition, we used a more elaborate method for clustering the entire curve of age-specific cervical cancer incidence using a Poisson-regression-based CEM clustering algorithm (lines 336-338, page 16). In the other publications, only age-standardized cervical cancer incidence or cervical cancer incidence in a certain age group were used.

Finally, the other publications provided less detailed description of their clustering/mapping steps. These steps were only reported briefly in supplementary material without intermediate results, whereas in this paper, we provided extensive details on each step with intermediate results, making them more reproducible and falsifiable.

To better explain the last two points we have added the following underlined part to the Discussion (lines 338-342, pages 16-17): “Secondly, we made use of a newly developed clustering method that is able to assess the similarities between cervical cancer incidence of different geographical units based on the entire age-specific pattern, instead of clustering on age-standardized cervical cancer incidence or cervical cancer incidence in a certain age group only. Finally, we provided a more detailed description of the clustering/mapping steps and intermediate results, which make them more reproducible and falsifiable.

Similarly, a more comprehensive explanation of the use of the sexual behaviour data is necessary; how did the authors decide which statistics to include in the methodology (e.g., age of first sexual intercourse, number of sexual partners lifetime)? Is there a guide to know which variables are ideal to apply in the technique?

We have included sexual behaviour variables that have previously been shown to be risk factors of HPV infection and cervical cancer risk, e.g., age of sexual debut and number of sexual partners [ref #26 Vaccerella 2006, ref #27 Schulte-Frohlinde 2021]. Furthermore, we used variables that are commonly available so that the analyses can be easily applied to other settings.

As far as we know, there is no established set of sexual behaviour variables for predicting the patterns of HPV prevalence and cervical cancer incidence. The good prediction performance in the India case study shows that using the selected set is sufficient. As sexual behaviour variables are highly correlated, including more variables might even risk overfitting.

To clarify these points we have included the following paragraph in the Discussion (lines 319-325, page 16): “In our analysis of classifying clusters of cervical cancer incidence, we only included some of the sexual behaviour variables available in the NACO report [15]. We selected variables that were previously shown to be risk factors of HPV infection and cervical cancer risk and that are commonly available so that the analyses can be easily applied to other settings, e.g., age of sexual debut and number of sexual partners [26, 27]. As far as we know, there is no established set of sexual behaviour variables for predicting the patterns of HPV prevalence and cervical cancer incidence. The good prediction performance shows that using the selected set is sufficient. As sexual behaviour variables are highly correlated, including more variables might even risk overfitting.”

Reviewer #1 (Recommendations for the authors):I have some general comments.The word "footprinting" is not very intuitive in the context of the manuscript. It sounds a bit strange, especially in the title; in the manuscript it's fine.

We have changed the title into ‘Approximating missing epidemiological data for cervical cancer through “footprinting”: a case study in India’ to explain the purpose of “footprinting”.

I have doubts about the utility of this framework when little data is available on HPV prevalence, sexual behavior, and cervical cancer incidence. What are the minimum data required to use this technique? What combinations of data types do you think we can do with this methodology (e.g., we only have sexual behavior but can use it with the HDI)?

While the proposed framework works better with more data, we think that it is flexible enough to be adapted to many cases with little data.

To answer the reviewer’s first question, there are three minimum data requirements. In our opinion, these requirements are reasonable and flexible enough to be fulfilled in many situations.

1) The first requirement is a data source of cervical cancer incidence, sexual behaviour, or HPV prevalence that has large enough, but not necessarily complete, coverage over all geographical units of interest. This data source, called “Pattern data” in the manuscript, is used to identify the main patterns of cervical cancer epidemiology.

2) A source of cervical cancer incidence, sexual behaviour, or HPV prevalence, or even alternative proxy as HDI and geographical location with coverage over the remaining geographical units to classify the unclustered geographical units to the identified clusters. In the manuscript, this data source is called “Footprint data”. This data source needs to cover some geographical units with identified clusters but not necessary all. Coverage on a part of the clustered geographical units should be sufficient for training the classifier used in the classification step.

3) Finally, data of cervical cancer incidence, sexual behaviour, and HPV prevalence for one geographical unit within each cluster are needed for the calibration of the projection models. However, these data do not necessarily need to come from the same geographical unit, as the data within a cluster should be similar enough.

In the reviewer’s example with only sexual behaviour and HDI, data requirement (3) is not fulfilled. Without any HPV prevalence and cervical cancer data at all, it would not be able to derive impact projections of cervical cancer intervention measures.

To answer reviewer’s second question, any combination of two of the three key cervical cancer epidemiological data, i.e., sexual behaviour, HPV prevalence, and cervical cancer incidence data, can serve as “Pattern” and “Footprint” data. As mentioned under data requirement (2) even proxies as HDI can be used as footprint data.

While we think our framework is flexible to be applied to many situation, we would like to stress that, as a general principle, we should not try overly approximate missing data when too little data are available to approximate from. This means that sometimes we might need to exclude from the analysis geographical units, or we might need to define bigger geographical units if the data are not granular enough. Only by doing so can we ensure the quality of the approximated data and the final impact projections.

Finally, it is worth noting that there are various widely recognized sources of databases of cervical cancer epidemiological data by country:

- cervical cancer incidence from CI5,

- HPV prevalence from ICO/IARC HPV information centre,

- sexual behaviour from the Demographic and Health Surveys (DHS) Program.

Therefore, there are likely ample data for application of the framework when considering countries as geographical units. For application within countries with states/provinces/municipalities as geographical units, data availability can differ from country to country.

To clarify these points, we have added the following paragraph to the Method (lines 144-163, pages 7-8): “For convenience of explanation, we assumed earlier that data availability occurs hierarchically. However, the framework can also be applied with less stringent data requirements. First, the source of Footprint data needs not necessarily cover all geographical units. It is still possible to train a classifier in the classification step with Footprint data available for only a part of clustered geographical units. Second, if none of the key cervical cancer epidemiological data (sexual behavior, HPV prevalence, and cervical cancer incidence data) have large enough coverage to serve as Footprint data, alternatives indicators of similarity, such as human development index and geographical distance, could also be used as substitute. However, the resulting classification performance might be suboptimal, as we expect these indicators to correlate less well with cervical cancer risk. Third, for the projection step, data of cervical cancer incidence, sexual behavior, and HPV prevalence needed for calibration of projection models need not necessarily belong to the same geographical unit. Calibration can be performed as long as the three types of data are available within each cluster.

With these less stringent data requirements, the proposed framework should sufficient flexible to be applied to many situations. However, one should still be cautious in applying the framework when there are little data. This means that, in some cases, we might need to exclude from the analysis some geographical units with too little data or redefine bigger geographical units if the data are not granular enough. Furthermore, we should assess the goodness-of-fit of the obtained clustering, performance of classification, correlation of data within different clusters, and calibration fits to ensure the validity of the final impact projections.”

Regarding sexual behavior data, how do you decide the statistics to include in the methodology (e.g., the median age of first age, number of sexual partners lifetime)? Is there a guide to know which variables are more recommended to apply in the technique?

A similar comment was raised by Reviewer #2. We have included sexual behaviour variables that have previously been shown to be risk factors of HPV infection and cervical cancer risk, e.g., age of sexual debut and number of sexual partners [ref #26 Vaccerella 2006, ref #27 Schulte-Frohlinde 2021]. Furthermore, we used variables that are commonly available so that the analyses can be easily applied to other settings.

As far as we know, there is no established set of sexual behaviour variables for predicting the patterns of HPV prevalence and cervical cancer incidence. The good prediction performance in the India case study shows that using the selected set is sufficient. As sexual behaviour variables are highly correlated, including more variables might even risk overfitting.

To clarify these points we have included the following paragraph in the Discussion (lines 319-325, page 16): “In our analysis of classifying clusters of cervical cancer incidence, we only included some of the sexual behaviour variables available in the NACO report [15]. We selected variables that were previously shown to be risk factors of HPV infection and cervical cancer risk and that are commonly available so that the analyses can be easily applied to other settings, e.g., age of sexual debut and number of sexual partners [26, 27]. As far as we know, there is no established set of sexual behaviour variables for predicting the patterns of HPV prevalence and cervical cancer incidence. The good prediction performance shows that using the selected set is sufficient. As sexual behaviour variables are highly correlated, including more variables might even risk overfitting.”

It will be interesting to validate this methodology in other regions with less available data to confirm its potential and utility.

We agree that it would be very interesting to validate this proposed methodology in other regions. Unfortunately, it was beyond the scope of this work. Currently, we are working on a project in which we try to apply footprinting to a collection of low- and middle-income countries.

I also have more specific comments or questions.When a cancer registry is available in both CI5C and NCDIR, and the information is not the same, which one do you get?

When a cancer registry is present in both CI5 and NCDIR, we take the data present in CI5. To clarify this point, we have added the following sentence to the “Data sources” section (lines 171-172, page 9): “When data of a registry is both reported in CI5 and NCDIR, we only used the data from CI5.”

Note that, in principle, the data in CI5 and NCDIR should be the same if they come from the same cancer registry. However, differences may arise when aggregation was done for different periods.

Can you explain how you decide the initial assignments in the CEM clustering algorithm (ranges of 100 repetitions)?

The initial assignments were randomly generated from a multinomial distribution. We added the underlined part to the Supplementary File (lines 34-35, page 3) to clarify this: “As different initial assignments could result in different final assignments, the above iterative process was repeated 100 times with different initial assignments, randomly generated from a multinomial distribution.”

Considering that none of the clusterings obtained using the CEM clustering algorithm are appropriate (2, 3, or 4), have you considered another method of clustering instead of combining clusters 2 and 3?

In the application considered, each Indian state can have multiple registries, while the sexual behaviour data were collected by Indian state. Hence, we needed to find a solution to deal with this. As the clustering obtained by combining clusters 2 and 3 already gave good separation for of high and low cervical cancer incidence, we did not consider it necessary to find alternative solutions.

Can you explain more extensively the differences between the previous approaches and your approach, with the strengths and limitations, and why this technique is more advantageous than the other ones?

To our knowledge, there are two publications on multi-country modelling for cervical cancer prevention with similar approaches (sometimes also called mapping) as the approach we proposed [ref #29 Brisson 2020. ref #30 Qendri 2020]. In these publications, similar data (sexual behaviour, cervical cancer data, and HPV prevalence) were used for clustering of countries. In our opinion (lines 329-334, page 16), an advantage of our approach is that we base our choice of key geographical units from which impact projections are extrapolated from on the pattern discovered in the data. Other approaches work with prefixed key geographical units because projection models have been calibrated to these geographical units in previous publications. However, no formal analyses have been done to show how well these key geographical units represent the cervical cancer epidemiological patterns across the geographical area of interest.

In addition, we used a more elaborate method for clustering the entire curve of age-specific cervical cancer incidence using a Poisson-regression-based CEM clustering algorithm (lines 336-338, page 16). In the other publications, only age-standardized cervical cancer incidence or cervical cancer incidence in a certain age group were used.

Finally, the other publications provided less detailed description of their clustering/mapping steps. These steps were only reported briefly in supplementary material without intermediate results, whereas in this paper, we provided extensive details on each step with intermediate results, making them more reproducible and falsifiable.

To better explain the last two points we have added the following underlined part to the Discussion (lines 338-342, pages 16-17): “Secondly, we made use of a newly developed clustering method that is able to assess the similarities between cervical cancer incidence of different geographical units based on the entire age-specific pattern, instead of clustering on age-standardized cervical cancer incidence or cervical cancer incidence in a certain age group only. Finally, we provided a more detailed description of the clustering/mapping steps and intermediate results, which make them more reproducible and falsifiable.

Reviewer #2 (Recommendations for the authors):The proposed framework's strength is difficult to evaluate because the steps and justification for the model variables were not clearly presented, nor were the models validated. Since the whole framework is built on one single imputation, how do the authors account for uncertainties about the estimation?

Uncertainty was accounted for in various steps of the framework. Firstly, the classification step was done through random forest (line 210, page 10), which summarizes the uncertainty of prediction by combining multiple classification trees. As we mentioned in the Method (lines 136-137, page 7), obtained classification probability can be used to weight projection outcomes. Furthermore, in the projection step, model calibration account for uncertainty of the target HPV prevalence by allowing model parameters that provide fit within the confidence intervals.

Perhaps the authors could consider validating these models by simulating models using data from countries where complete datasets are available (e.g., the US).

We agree that it would be very interesting to validate this proposed methodology in other regions. Unfortunately, it was beyond the scope of this work. Currently, we are working on a project in which we try to apply footprinting to a collection of low- and middle-income countries.

The manuscript would be strengthened by a more explicit stepwise delimitation of how to apply this model to data.

We acknowledge that the framework could be more clearly presented and have added additional explanation in the following places to do so:

– Concerning the framework steps, in Method (144-163, pages 7-8): “For convenience of explanation, we assumed earlier that data availability occurs hierarchically. However, the framework can also be applied with less stringent data requirements. First, the source of Footprint data needs not necessarily cover all geographical units. It is still possible to train a classifier in the classification step with Footprint data available for only a part of clustered geographical units. Second, if none of the key cervical cancer epidemiological data (sexual behavior, HPV prevalence, and cervical cancer incidence data) have large enough coverage to serve as Footprint data, alternatives indicators of similarity, such as human development index and geographical distance, could also be used as substitute. However, the resulting classification performance might be suboptimal, as we expect these indicators to correlate less well with cervical cancer risk. Third, for the projection step, data of cervical cancer incidence, sexual behavior, and HPV prevalence needed for calibration of projection models need not necessarily belong to the same geographical unit. Calibration can be performed as long as the three types of data are available within each cluster.

With these less stringent data requirements, the proposed framework should sufficient flexible to be applied to many situations. However, one should still be cautious in applying the framework when there are little data. This means that, in some cases, we might need to exclude from the analysis some geographical units with too little data or redefine bigger geographical units if the data are not granular enough. Furthermore, we should assess the goodness-of-fit of the obtained clustering, performance of classification, correlation of data within different clusters, and calibration fits to ensure the validity of the final impact projections.”

– Concerning selection of model variables (lines 319-325, page 16): “In our analysis of classifying clusters of cervical cancer incidence, we only included some of the sexual behaviour variables available in the NACO report [15]. We selected variables that were previously shown to be risk factors of HPV infection and cervical cancer risk and that are commonly available (e.g., age of sexual debut and number of sexual partners) so that the analyses can be easily applied to other settings [26, 27]. In the India case study, the good classification performance shows that using the selected set is sufficient. As sexual behaviour variables are highly correlated, including more variables might even risk overfitting.”

The paper would be strengthened by including evidence for the utility of estimating HPV and cervical cancer rates based on sexual behaviours.

We have included sexual behaviour variables that have previously been shown to be risk factors of HPV infection and cervical cancer risk, e.g., age of sexual debut and number of sexual partners [ref #26 Vaccerella 2006, ref #27 Schulte-Frohlinde 2021]. Furthermore, we used variables that are commonly available so that the analyses can be easily applied to other settings.

As far as we know, there is no established set of sexual behaviour variables for predicting the patterns of HPV prevalence and cervical cancer incidence. The good prediction performance in the India case study shows that using the selected set is sufficient. As sexual behaviour variables are highly correlated, including more variables might even risk overfitting.

To clarify these points we have included the following paragraph in the Discussion (lines 319-325, page 16): “In our analysis of classifying clusters of cervical cancer incidence, we only included some of the sexual behaviour variables available in the NACO report [15]. We selected variables that were previously shown to be risk factors of HPV infection and cervical cancer risk and that are commonly available so that the analyses can be easily applied to other settings, e.g., age of sexual debut and number of sexual partners [26, 27]. As far as we know, there is no established set of sexual behaviour variables for predicting the patterns of HPV prevalence and cervical cancer incidence. The good prediction performance shows that using the selected set is sufficient. As sexual behaviour variables are highly correlated, including more variables might even risk overfitting.”

Lastly, it seems that the impact assessment of this work has already been published. Why was the current manuscript sent for publication after the paper on impact assessment was published?

This manuscript was posted on MedRxiv at the same time as we submitted the impact assessment paper. The other one was accepted more quickly.